# Allele-Specific CG/CCWGG Methylation of the PSA Promoter Discriminates Aggressive, Indolent, and Benign Prostate Cell Lines and Is Involved in the Regulation of PSA Expression

**DOI:** 10.3390/ijms26031243

**Published:** 2025-01-31

**Authors:** Mikhail Baryshev, Egils Vjaters

**Affiliations:** 1Institute of Microbiology and Virology, Riga Stradins University, Ratsupites Str 5, LV-1067 Riga, Latvia; 2Institute of Oncology and Molecular Genetics, Riga Stradins University, Pilsoņu Str 13, LV-1007 Riga, Latvia; egils.vjaters@rsu.lv

**Keywords:** allele-specific methylation, CpG, CCWGG methylation, DNA methylation, prostate cancer, PSA

## Abstract

Prostate-specific antigen remains a cornerstone biomarker for prostate cancer diagnosis and management. However, the molecular mechanisms regulating its expression, particularly through DNA methylation, are not fully understood. Here, we report a comprehensive analysis of allele-specific CpG and CCWGG methylation in the proximal PSA promoter across aggressive (PC3), indolent (LNCaP), benign (BPH1), and normal (HPrEpiC) prostate cell lines and provide insights into the unique methylation patterns associated with these states. Our findings reveal that PC3 cells, representing an aggressive PCa phenotype, exhibit complete biallelic methylation of the PSA promoter, leading to PSA gene silencing. Conversely, LNCaP cells display a fully unmethylated promoter with biallelic PSA expression. Interestingly, BPH1 cells display a monoallelic CG/CCWGG methylation pattern, yet fail to express PSA, suggesting imprinting defects or RNA decay mechanisms. Notably, acquisition of biallelic PSA promoter methylation status in PC3 was accompanied by upregulation of DNMT1, whereas unmethylated PSA promoter state in LNCaP was associated with downregulation of DNMT1. These findings highlight distinct methylation patterns in the PSA promoter that differentiate between aggressive, indolent, and benign prostate states. Translating this epigenetic insight into clinical diagnostics could enhance the precision of PSA-based diagnostics, addressing limitations such as false negatives in PSA testing for aggressive PCa. Further exploration of CCWGG methylation’s role in imprinting and monoallelic expression is warranted, particularly in patient-derived samples.

## 1. Introduction

For a long time, PSA has been used as an established tumor marker for prostate adenocarcinoma in clinical practice for the diagnosis of prostate cancer (PCa) [1,2,3]. Originally identified as a tissue-specific protein exclusively expressed by prostate epithelial cells, PSA has also been surprisingly found in a subset of breast tumors [4]. Despite the generally accepted fact that PSA is organ-specific but not cancer-specific, it retains diagnostic value for cancer detection and, as we demonstrate, there is a solid basis for this [5]. PSA levels may be elevated in benign prostatic hyperplasia (BPH), prostatitis, and other nonmalignant conditions in proportion to the increase in cell mass. Based on our findings, PSA levels may reflect cancer when there is biallelic PSA expression, as opposed to monoallelic expression, which we suggest occurs in normal or benign tissues. PSA expression is regulated by genetic elements and epigenetic factors involved in gene expression. While exploring the altered status of DNA methylation as a disease-specific biomarker, some researchers focus primarily on CpG islands (CGIs) and overlook the methylation of individual CpGs located in proximal promoters [6]. However, it has been shown that methylation of even a single CpG site near the transcription start site (TSS) of *PPARγ2*, a key regulator of adipogenesis, can silence the gene and block the adipocyte differentiation pathway [7,8]. Given the clinical utility of the PSA gene in PCa testing, we studied the methylation of CpG and CCWGG (W = A/T) sites in the proximal promoter of PSA in both PSA-expressing and non-expressing cancer cell lines (LNCaP and PC3, respectively), as well as in noncancerous BPH1 and HPrEpiC cells. Additionally, PA1 cells, which are of nonprostatic origin and lack PSA expression, were included in the analysis.

In this study, we demonstrate that the methylation status of the non-CpG island PSA promoter exhibits a distinctive methylation pattern in a PCa cell line model. This pattern ranges from a complete lack of methylation in LNCaP cells, to a monoallelic type combined with methylation of two proximal CCWGG sites in BPH1 cells, and finally to complete biallelic 5-methylcytosine modification of CpGs in PC3 cells. The monoallelic distribution of CG/CCWGG in BPH1 and PA1 cells, the latter of which are nonpermissive for PSA expression, suggests the possibility of PSA imprinting or monoallelic regulation of gene expression. The fact that the two proximal CCWGG motifs of the PSA promoter are located near the TATA box and TSS confirms the importance of CCWGG methylation in these cell lines and suggests the involvement of this methylation pattern in the regulation of gene expression. To achieve single base-pair resolution of CpG and CCWGG methylation, we utilized bisulfite-treated DNA (BTD).

## 2. Results

### 2.1. The Proximal PSA Promoter Has an Individual CG/CCWGG Methylation Pattern in Prostate Cell Lines

#### 2.1.1. The Monoallelic Methylation of CG/CCWGG Sites in the PSA Promoter Observed in BPH1 Cells and the Biallelic Stochastic Distribution of CpG Methylation in HPrEpiC Cells Represent Distinctive Patterns Specific to These Cell Lines

The PSA promoter, spanning nucleotide positions −393 to +51 and containing six CpGs and five CCWGG epigenetic marks (Figure 1a–c), which are targets of DNMTs and may epigenetically influence gene expression, was subjected to allelic methylation analysis for its status. BTD DNA was used to amplify 443 bp of the PSA proximal promoter with a set of bisulfite-specific primers (Table 1) for CG/CCWGG methylation analysis. The methylation status of the PSA promoter was assessed using a cloning and sequencing approach in prostate cancerous and noncancerous cell lines. Twenty clones were analyzed, representing a mixture of PSA promoter fragments derived from parental alleles. The G-158A SNP, located in the AREI region of the PSA promoter (Figure 1c), is associated with disease severity and levels of circulating tumor cells [9]. This SNP was considered in the genotyping of prostate cell lines for allele-specific methylation analysis. Sequencing of 20 clones revealed the following genotypes for SNP G-158A: HPrEpiC (A/G), BPH1 (A/G), LNCaP (A/A), PC3 (A/A), and PA1 (A/A).

In the HPrEpiC cell line, stochastic CpG methylation of the PSA promoter region was detected, with both alleles methylated across all 16 clones (Figure 1d, Appendix A). In contrast, the BPH1 cell line exhibited CpG methylation alongside methylation of two proximal CCWGG pentanucleotides (CCWGG)_2_ on the same allele (Figure 1d). Analysis of the clones showed that half were fully methylated, while the other half were unmethylated. Notably, eight unmethylated clones corresponded to the A allele, while five clones with CG/CCWGG methylation corresponded to the G allele. Three A allele clones showed only CG methylation. We observed that the methylation of the proximal (CCWGG)_2_ sites on the same allele consistently coexisted with CpG methylation. Interestingly, the PA1 cell line, which does not express PSA, exhibited a similar methylation pattern of CpG and CCWGG sites (Figure 1d). We hypothesize that the active allele in PA1 cells is silenced by DNA methylation, potentially due to imprinting, while in BPH1 cells, improper imprinting leads to silencing of the active allele, resulting in loss of PSA expression. We also cannot rule out silencing via noncanonical imprinting mechanisms. Given the strict methylation of the proximal (CCWGG)_2_ sites in PA1 and BPH1 cells, we propose that these pentanucleotide sites may represent a novel epigenetic mark functionality involved in regulating monoallelic gene expression. The absence of PSA expression in the HPrEpiC cell line, which exhibits a stochastic biallelic CG methylation distribution but lacks CCWGG methylation, further supports the potential role of CCWGG motif methylation in regulating monoallelic PSA expression. In addition, in the 5′ flanking region of the PSA gene, beyond the enhancer and proximal promoter, we identified two clustered CpG/CCWGG regions: A—carrying six CpG and five CCWGG motifs—and B—having eleven CpG and four CCWGG sites. (Figure 1a). Whether these loci, along with the CG/CCWGG sites in the PSA promoter, function as germline differentially methylated regions (gDMRs) or CCWGG methylation acts as a secondary imprint remains to be clarified.

#### 2.1.2. The Biallelic Methylation of the PSA Promoter in PC3 Cells and the Fully Unmethylated State in LNCaP Cells Represent Cancer-Specific Methylation Patterns

PC3 and LNCaP cancer cell lines exhibit an inverse methylation pattern in the PSA promoter: biallelic methylation in PC3 and complete unmethylation in LNCaP (Figure 2, Appendix A). PC3 cells, representing an aggressive PCa phenotype, do not express PSA due to the silencing of both alleles through methylation. Clinically, this gene silencing could result in false-negative outcomes in PSA-based PCa testing, as cancer cells continue to proliferate without a corresponding increase in PSA levels.

Some CCWGG methylation was observed in PC3 cells, including methylation of proximal (CCWGG)_2_ motifs detected in a single clone (Figure 2, Appendix A). In contrast, the LNCaP cell line, which represents a less aggressive, indolent PCa phenotype, expresses PSA and shows a methylation-free PSA promoter, devoid of both CpG and CCWGG methylation (Figure 2). Thus, these two prostate cancer cell lines exhibit distinct, PCa-specific PSA promoter methylation statuses, potentially linked to a loss of imprinting (LOI) if monoallelic PSA expression regulation by imprinting is assumed.

### 2.2. LNCaP Cells Biallelically Express PSA mRNA

According to the literature, neither the cancerous PC3 nor the benign BPH1 cell lines express PSA mRNA, unlike the LNCaP cell line [10]. Due to the lack of data on PSA expression in HPrEpiC cells, we used RT-PCR with a “40 plus 40” cycle nested RT-PCR protocol, which is sensitive enough to detect low levels of PSA cDNA in reverse-transcribed RNA samples [11]. PA1, which is nonpermissive for PSA expression, along with PC3, LNCaP, and BPH1, were analyzed for PSA mRNA expression. PSMA expression was also assessed to confirm the phenotypic “fingerprint” of the cell lines. None of the cell lines, except for LNCaP, expressed PSA in any RT or nested RT-PCR analysis (Figure 3a,b). Although HPrEpiC is a normal cell line, it does not express PSA mRNA (Figure 3a), but it does express PSMA mRNA, albeit at lower levels than LNCaP, consistent with the differential PSMA expression between prostate cancer and normal tissue (Figure 3a). Interestingly, the noncancerous prostate cell line PNT2 does not express PSA [12]. Allele-linked single-nucleotide polymorphisms (SNPs) are useful features to estimate allelic expression in heterozygous genomes. We employed an RT-PCR cloning/sequencing approach to detect polymorphisms in the transcribed region and assess PSA allele expression. The split peak in the chromatogram generated by sequencing the ORF PSA RT-PCR products indicates the presence of heterozygosity in the PSA alleles (Figure 3c). The same PSA polymorphism was detected in the amplification of two independent cDNA samples. The GA polymorphism identified at nucleotide 447 in exon 3 was further confirmed by sequencing independent clones. Sequencing of 20 clones of the full-length cDNA of PSA transcript variant 1 revealed the presence of mRNA with G and A nucleotides distributed across 10 clones of each nucleotide variant, as shown in Figure 3d, indicating biallelic expression of PSA.

### 2.3. Allele-Specific CG/CCWGG Methylation of the PSA Promoter Is Involved in the Regulation of PSA Expression

Comparative analysis of PSA mRNA expression, in relation to allelic CG/CCWGG methylation status in prostate cell lines, revealed that neither PC3 cells with complete biallelic CpG methylation nor normal HPrEpiC cells with heavily methylated biallelic CpG status (Figure 1d) express PSA (Figure 3a). This lack of expression is attributed to the well-documented repressive role of CpG methylation in gene silencing. Intriguingly, the allelic CG/CCWGG methylation status in PA1 cells exhibited an exclusively monoallelic arrangement, with one allele completely unmethylated, while the other was fully CpG methylated and included proximal (CCWGG)_2_ motifs (Figure 1d). However, despite this pattern, PA1 cells are nonpermissive for PSA expression, and the absence of PSA mRNA suggests the involvement of additional regulatory mechanisms, such as genomic imprinting, in controlling monoallelic expression.

To our surprise, the BPH1 cell line, which exhibits a monoallelic CG/CCWGG methylation profile resembling that of PA1 (Figure 1d), also does not express PSA (Figure 3a). Whether the inability of BPH1 cells to express PSA is due to a failure to imprint or is caused by a nonsense-mediated RNA decay mechanism clearing PSA mRNA remains to be elucidated. However, the LNCaP cell line, which exhibits biallelic PSA expression and lacks both CpG and proximal (CCWGG)₂ methylation, clearly demonstrates the involvement of allelic CG/CCWGG methylation in the regulation of PSA expression.

### 2.4. DNMT1 and DNMT3B Exhibit Overall Expression Profiles in BPH1 and PA1 Cells But Are Differentially Expressed in LNCaP, PC3, and HPrEpiC Cell Lines

The mRNA expression of the three DNMTs was examined by RT-qPCR in a prostate cancer cell line model with the primers listed in Table 1. DNMT1 and DNMT3B are equally expressed in BPH1 cells, and their expression profile is similar to that in PA1 cells, where monoallelic CG/CCWGG methylation of the PSA promoter occurs (Figure 4a). The level of DNMT1, which is responsible for maintaining established methylation patterns, is significantly reduced (by 0.3-fold) in the LNCaP cell line, where the PSA promoter is completely unmethylated. In contrast, the fully methylated PSA promoter in PC3 cells is associated with a 1.2-fold increase in DNMT1 expression (Figure 4a). Additionally, transcripts of DNMT3B, a de novo DNA methyltransferase, are slightly downregulated in LNCaP cells, reduced in PC3 cells, and heavily reduced in HPrEpiC cells, which avoid CCWGG methylation. The expression profile of DNMTs in normal prostate tissue (Figure 4b) shows higher expression of DNMT1, intermediate expression of DNMT3A, and lower expression of DNMT3B, suggesting that all studied cell lines exhibit an altered pattern of DNMT expression compared to normal tissues [13].

Previous studies have shown that loss of DNMT3A or DNMT3B can result in the loss of DNA methylation in specific loci of imprinted genes such as IGF2 and XIST [14,15]. Additionally, DNMT3B has been implicated in the regulation of stochastic and monoallelic expression of Pcdh isoforms [16]. While DNMT3A is expressed at a relatively high level in all cell lines and likely does not influence the CpG/CCWGG methylation pattern in the PSA promoter, an imbalance between DNMT1 and DNMT3B observed in the PC3 and LNCaP cell lines may lead to the loss of the PSA-specific methylation pattern. There also appears to be dysregulation of the DNMT expression pattern in HPrEpiC cells, which exhibit stochastic biallelic PSA promoter methylation.

## 3. Discussion

Although PSA is produced by normal, hyperplastic, and neoplastic epithelial cells, several prostate cancer cell lines commonly used in PCa research—such as PC3 and Du145—and benign prostatic hyperplasia (BPH1) cells, as well as HPrEpiC cells, which are considered normal, do not produce PSA. In this study, we identified proximal PSA promoter methylation patterns in a PCa cell line model, focusing on CpG and CCWGG motifs, and report distinct PSA promoter methylation profiles. These include cancer-specific patterns (Figure 2) and others that reflect a noncancerous state (Figure 1d). LNCaP cells, which have a PSA promoter completely free of CpG and CCWGG methylation on both alleles, show biallelic PSA expression (Figure 3d). In contrast, PC3 cells lack PSA expression due to complete biallelic promoter methylation. At the same time, BPH1 cells, which exhibit monoallelic methylation of the PSA promoter (Figure 1d), do not express PSA (Figure 3a). Allele-specific CpG/CCWGG methylation was observed in both PA1 and BPH1 cells, suggesting that PSA expression could be regulated through mechanisms such as imprinting or random monoallelic expression. It is well established that monoallelic gene methylation is a prerequisite for monoallelic gene expression [17]. However, the monoallelic methylation observed in BPH1 and PA1 cells, coupled with their inability to express PSA mRNA, raises a critical question: does the failure to imprint (i.e., silencing of the active allele) occur in BPH1 cells, or is the RNA degraded through a nonstop decay mechanism? This would imply that proper imprinting is maintained in PA1 cells. Resolving this issue requires further investigation. Notably, the possibility of “noncanonical” imprinting defects—those independent of DNA methylation—cannot be excluded.

The molecular mechanisms underlying the establishment and maintenance of autosomal monoallelic expression (MAE) in mammals remain largely unknown. A recently published study demonstrated that DNA methylation plays a crucial role in maintaining allelic methylation imbalances at various loci [18]. Parent-of-origin-specific DNA methylation, known as “canonical imprinting”, is established during gametogenesis when maternal and paternal genomes are separated and independently epigenetically modified. This process creates inherited imprinting markers that directly or indirectly regulate the expression of imprinted genes.

Unlike germline differentially methylated regions, imprinted secondary DMRs (sDMRs) acquire allele-specific DNA methylation during embryonic development, rather than inheriting it from germ cells. Although sDMRs may contribute to the maintenance of imprinting, their role remains untested in most genomic regions [19].

Based on our findings of a monoallelic CpG/CCWGG methylation pattern, we hypothesize that this pentanucleotide mark may play a role in imprinting and/or MAE. Whether CCWGG methylation is established during gametogenesis or represents sDMRs acquired during embryonic development remains an open question that warrants further investigation. Although no specific DNMTs have been identified for the CCWGG substrate, evidence for this type of methylation is well documented [20,21,22,23]. The functional impact of CCWGG methylation on gene expression is controversial. It is suggested to range from repressive to promotive effects, potentially forming distinctive patterns that enable adaptation to altered environmental conditions. This functionality may depend on the context of locus-specific methylation and the presence of methylation mark readers [22,24]. In this study, we demonstrate that CG/CCWGG dual epigenetic marks are involved in the regulation of PSA expression. However, the extent and spectrum of CCWGG functionality in PSA expression regulation remain to be elucidated. Specifically, whether CCWGG is linked to expression repression or activation or plays a role in specifying which allele is methylated requires further investigation.

Interestingly, if PSA is imprinted and monoallelically expressed in normal cells, both the PC3 and LNCaP cell lines can be viewed as exhibiting loss of imprinting (LOI). LOI is a state of dysregulated gene expression often associated with pathological conditions, as depicted in Figure 5 [25,26]. For instance, the biallelic expression of the insulin-like growth factor-2 (IGF2) gene is known to promote oncogenesis by inhibiting apoptosis [27]. Similarly, LOI resulting in biallelic silencing of genes like PEG3, P57, and IGF2R has been observed in oligodendrogliomas, breast cancer, and hepatocellular carcinomas, respectively [28,29,30]. Moreover, some studies report PSA expression in nonprostatic cells [4]. Analysis of PSA expression in 33 TCGA cancer types revealed low levels in adrenocortical carcinoma (0.3 GPM), colon adenocarcinoma (0.2 GPM), kidney chromophobe carcinoma (8.1 GPM), and rectum adenocarcinoma (0.3 GPM). Similarly, normal tissues from GTEx RNA-seq data also showed detectable PSA expression (Appendix A). According to our findings, this could be explained by LOI arising from the loss of methylation status in both alleles on one hand and the biallelic promoter methylation state on the other hand, as shown in Figure 5. This is consistent with other imprinted genes where LOI is associated with the occurrence of a pathological condition.

Despite numerous studies linking gene methylation to PCa, no methylation-based biomarkers have been clinically adopted. Our study of PCa cell lines suggests a potential diagnostic application of CG/CCWGG methylation in the PSA promoter. Specifically, an elevated PSA level with monoallelic promoter methylation could indicate benign prostatic hyperplasia or other nonmalignant conditions. Conversely, a significantly elevated PSA level due to biallelic PSA expression, accompanied by an unmethylated promoter, would suggest PCa.

Importantly, a low PSA level may reflect the presence of aggressive, PSA non-secreting PCa phenotypes, such as PC3 cells with biallelic PSA promoter methylation. This scenario highlights a clinical risk: biallelic methylation-mediated PSA silencing in aggressive PCa could result in false-negative PSA tests. This issue aligns with reports that many men harbor PCa despite low serum PSA levels [31]. Thus, detection of biallelic PSA promoter methylation even at low PSA levels in patients with other features of prostate cancer may serve as an indicator of aggressive PCa, potentially increasing diagnostic accuracy.

It should be emphasized that some factors such as infections, inflammation, stress or even environmental factors can influence CCWGG methylation through complex mechanisms involving DNMT regulation, oxidative stress and cytokine signaling. Mainly, these factors can influence CCWGG methylation leading to hypomethylation in repetitive or regulatory regions enriched in CCWGG motifs. CCWGG motifs are known to be enriched in transposable elements that can become hypomethylated during inflammation or stress leading to their activation and genomic instability. The observed proximal methylation of (CCWGG)2 in both PA1 and BPH1 cell lines suggests that this DNA methylation is highly specific. Since PA1 cell line is derived from human ovarian teratocarcinoma and has a female karyotype (XX), while BPH1 cell line is of prostatic origin and has the same methylation pattern, we are inclined to think that proximal methylation of (CCWGG)2 in the PSA promoter is cell type specific. Overall, the observed CCWGG methylation highlights the importance of an integrative study of CCWGG methylation dynamics considering factors influencing DNA methylation

By profiling CpG/CCWGG methylation in the PSA promoter across a PCa cell line model, we present novel data that link PSA promoter methylation to specific disease states. Our findings have the potential to refine PSA-based diagnostic strategies by correlating serum PSA levels with distinct methylation patterns, thereby enhancing diagnostic precision and providing insights into the regulatory mechanisms of PSA expression. We propose that the CCWGG motif may act as a subtle allele-specific methylation signal, potentially representing a novel regulatory mark. This mark could play a role in allele-specific methylation mechanisms and/or imprinting, warranting further investigation into its impact on monoallelic gene expression. Future studies should validate these findings in patient-derived samples to confirm their clinical relevance.

According to our data, assessing allele-specific methylation of the PSA promoter can offer a more nuanced understanding of disease-specific states. This approach could significantly improve the accuracy of PSA-based testing for PCa in clinic practice, offering a way to more personalized and effective diagnostic tools, especially in patients harboring aggressive prostate cancer, even with low PSA levels or a negative MRI results.

## 4. Materials and Methods

### 4.1. Cell Lines

Human prostatic carcinoma cell lines PC3 (ATCC^®^ CRL1435™) and LNCaP (ATCC CRL-1740™) were purchased from the American Type Culture Collection (ATCC); BPH1 (DSMZ no.: ACC 143) came from the DSMZ, German Collection of Microorganisms and Cell Cultures; and GmbH and the human prostatic epithelial cell line (HPrEpiC) were obtained from the ScienCell Research Laboratories (Walkersville, MD, USA). All cell lines were cultured at 37 °C in a humidified incubator with 5% CO_2_ according to the manufacturer’s protocol. Other chemicals and reagents were purchased from Sigma-Aldrich (Spruce St., Saint Louis, MO, USA).

### 4.2. DNA Extraction and Bisulfite Treatment

Genomic DNA was obtained by incubating cells overnight in TES buffer containing 0.1% SDS and 100 µg/mL proteinase K at 55 °C, followed by phenol/chloroform extraction and isopropanol precipitation. Two micrograms of DNA, dissolved in 50 µL of TE buffer, were denatured in 0.3 M NaOH at 37 °C for 15 min. The denatured DNA was then mixed with 550 µL of a freshly prepared solution containing 10 mM hydroquinone and 3 M sodium bisulfite (pH 5.0) and incubated under mineral oil at 50 °C for 12 h. After bisulfite treatment, the DNA was desalted by isopropanol precipitation, desulfonated with 0.3 M NaOH at room temperature for 5 min, and then precipitated with ethanol. The converted DNA was dissolved in 100 µL of TE buffer and stored at −20 °C. For each PCR reaction, 3 µL of the precipitated DNA were used.

### 4.3. Bisulfite Sequencing Analysis

Bisulfite-treated DNA (BTD) was used to amplify the proximal region of the PSA promoter with primers specific for bisulphite-converted DNA listed in Table 1. The PCR was carried out using 2.5 units of homemade Taq polymerase in a final volume of 50 μL and the following cycling conditions: 5 min at 95 °C, followed by 35 cycles (30 s denaturation at 95 °C, annealing for 30 s at 62 °C, and elongation at 72 °C for 1 min). The PCR products were gel purified and cloned using a TOPO TA cloning kit. To prevent clonal amplification of sequences, the competent cells transformed were plated immediately after heat shock, excluding shaking bacteria for 1 h. Plasmid DNA of individual clones was purified with the DM method [32]. The recombinant plasmids were sequenced in both directions using M13 forward and reverse primers (Invitrogen) and an ABI BigDye Terminator Cycle Sequencing Kit v3.1 (Thermo Fisher, Waltham, MA, USA) with a Gene Amp 9700 PCR System (Thermo Fisher, Carlsbad, CA, USA). The sequences were detected with an ABI 3130XL Genetic Analyzer (Applied Biosystems, Foster City, CA, USA). Promoter methylation analysis was performed by aligning sequenced clones with the PSA promoter region, where cytosine to uracil was converted in an in silico experiment. The efficiency of cytosine to uracil conversion was estimated as the ratio of cytosine in a non-CpG context to the total number of cytosines in the region. The clones with an efficiency of cytosine conversion less than 98% were omitted from the analysis.

### 4.4. RNA Extraction

Total RNA was extracted from HPrEpiC, BPH-1, LNCaP, and PC3 cells using TRIzol reagent (Invitrogen, Carlsbad, CA, USA) following the supplier’s instructions. The extracted RNA was dissolved at 55 °C in sterile water treated with diethylpyrocarbonate (DEPC) (Sigma, St. Louis, MO, USA). The concentration of the total RNA was measured with a Qubit Fluorometer (Invitrogen, Carlsbad, CA, USA), and RNA integrity was evaluated through 1% agarose gel electrophoresis to visualize a certain banding pattern and by amplification of the housekeeping gene. The resulting PCR product was subjected to gel electrophoresis in 1% agarose and visualized by ethidium bromide staining and UV transillumination.

### 4.5. cDNA Synthesis, RT/Nested RT-PCR

Five micrograms were reverse transcribed into cDNA using the Maxima H Minus First Strand cDNA synthesis kit with ds DNase (Thermo Fisher Scientific Inc., Waltham, MA, USA) according to the manufacturer’s instructions. PSA and PSMA expression as specific markers of prostate cell lines was estimated by RT-PCR. Two microliters of the resulting PSA PCR product were subjected to a second nested PCR using the primers listed in Table 1. Each cDNA sample was assayed in duplicate using primers for both PSA and PSMA. The fidelity of amplification for the nested primer products was confirmed by sequencing and was found to be consistent within one base pair of the PSA reference sequences.

### 4.6. Quantitative Real-Time PCR

To quantify the level of DNMT expression, cDNA was amplified with the CFX96™ Real-Time PCR detection system (Bio-Rad Laboratories Inc., Richmond, CA, USA) and PerfeCTa SYBR Green FastMix (Quanta BioSciences Inc., Beverly, MA, USA). Built-in data analysis modules with automatic baseline subtraction and a threshold setting of CFX manager were used to analyze the normalized (ΔΔCT) DNMT mRNA expression. TATA box-binding protein (TBP), the housekeeping gene, was used to obtain relative normalized mRNA expression. Melting curve analysis was performed, ramping from 60 °C to 90 °C and rising by 0.5 °C every 2 s. The relative expression levels were calculated using the 2^−ΔΔCT^ method and CFX Maestro software version 2.3. All experiments were conducted in triplicate.

### 4.7. Primer Design

The primer-BLAST software tool (CFX Maestro Software 2.3) was used to design new target-specific primers for RT–PCR experiments. This tool is available at http://www.ncbi.nlm.nih.gov/tools/primer-blast (accessed on 20 May 2024). All amplicon primers were designed to span exon-exon boundaries, preventing the amplification of genomic DNA. The specificity of the amplicons and primer pairs was verified in silico using BLAST alignment tools from the National Center for Biotechnology Information (NCBI), accessed on 12 February 2023.

For bisulfite-modified DNA, primer design required adherence to a specific rule: since the DNA strands are no longer complementary after bisulfite treatment, each primer set amplifies only one strand of the target sequence. The first primer was designed to anneal to the converted target sequence, while the second primer was designed to anneal to the extension product of the first primer, rather than the opposite template strand.

Sequence Manipulation Suite: PCR Primer Stats were used to generate robust primers for bisulfite PCR sequencing analysis. These modules are available online at https://www.bioinformatics.org/sms2/pcr_primer_stats.html (accessed on 20 February 2024).

### 4.8. Bioinformatics

National Centre for Biotechnology Information (NCBI) database resources for bioinformatics were used to assess DNMT expression in normal prostate tissue. The Basic Local Alignment Search Tool (BLAST) commonly used in bioinformatics was applied to search for similarities and identify homologous sequences. The Genotype-Tissue Expression (GTEx) project data of gene expression in 54 tissues from GTEx RNA-seq of 17,382 samples and 948 donors (V8, August 2019) and the RNA expression level in 33 TCGA cancer tissues (GENCODE v23) produced by the consortium were used with the UCSC Human Genome Browser (GRCh38/hg38). The TCGA chose cancers for study based on two broad criteria: poor prognosis/overall public health impact and availability of human tumor and matched normal tissue samples that meet TCGA standards.

### 4.9. Statistical Methods and Associated Software

The CFX96™ Real-Time PCR detection system (CFX Maestro Software 2.3) associated software and algorithm were used in this study.

## Figures and Tables

**Figure 1 ijms-26-01243-f001:**
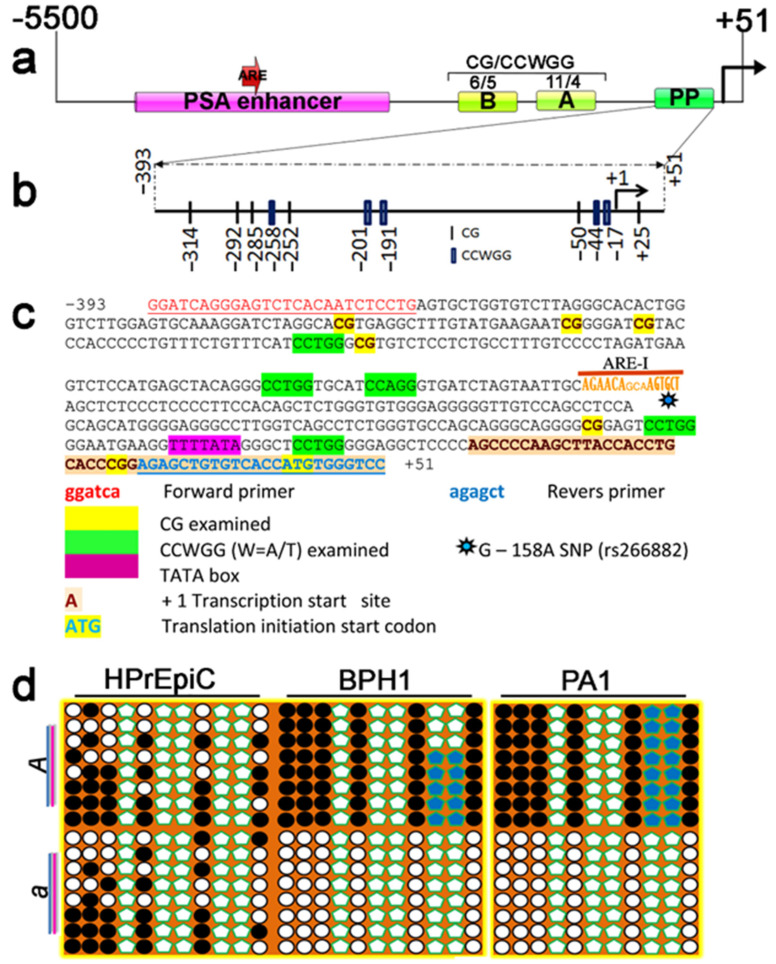
The proximal PSA promoter has an individual CG/CCWGG methylation pattern in prostate cell lines. (**a**) Schematic illustration of the 5’ flanking region of the PSA gene. The proximal promoter (PP) and the CG/CCWGG A and B regions are shown. (**b**) Schematic of the proximal PSA promoter; short thin vertical lines indicate the position of the CG; short wide vertical lines represent CCWGG positions. An arrow indicates the transcription start site (TSS) (+1 bp). (**c**) Distribution of CpG and CCWGG, W = A/T) sites within the PP of the genomic sequence kallikrein-related peptidase 3 KLK3/PSA (source: HGNC Symbol; Acc: HGNC: 6364; Chromosome 19: 50,854,915–50,860,764 forward strand). (**d**) Bisulfite sequencing analyses of the PSA PP. The status of PSA methylation in prostate cell lines and cells of nonprostatic origin was analyzed. *A* and *a* represent the parental alleles. At the bottom of the illustrations, the methylation status of the CpGs is shown.

**Figure 2 ijms-26-01243-f002:**
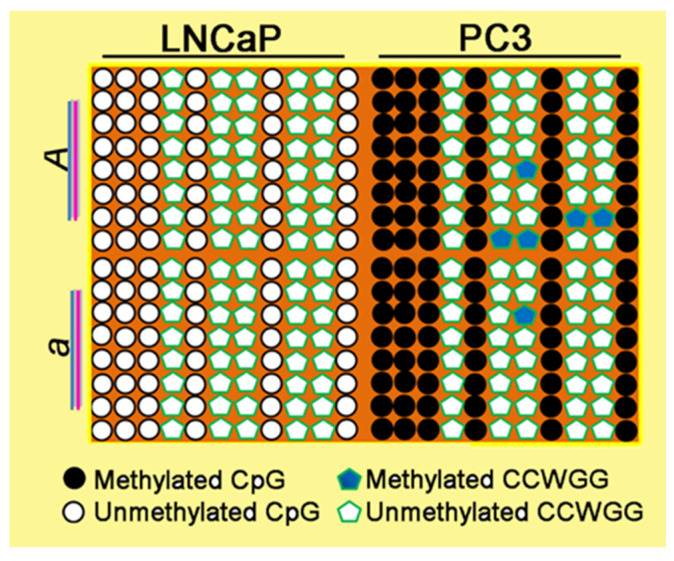
The methylated in PC3 and unmethylated in LNCaP cells proximal PSA promoter is cancer specific. Bisulfite sequencing analyses of the PSA PP in PC3 and LNCaP cells. *A* and *a* represent the parental alleles. At the bottom of the illustrations, the methylation status of the CpGs and CCWGG motif is shown.

**Figure 3 ijms-26-01243-f003:**
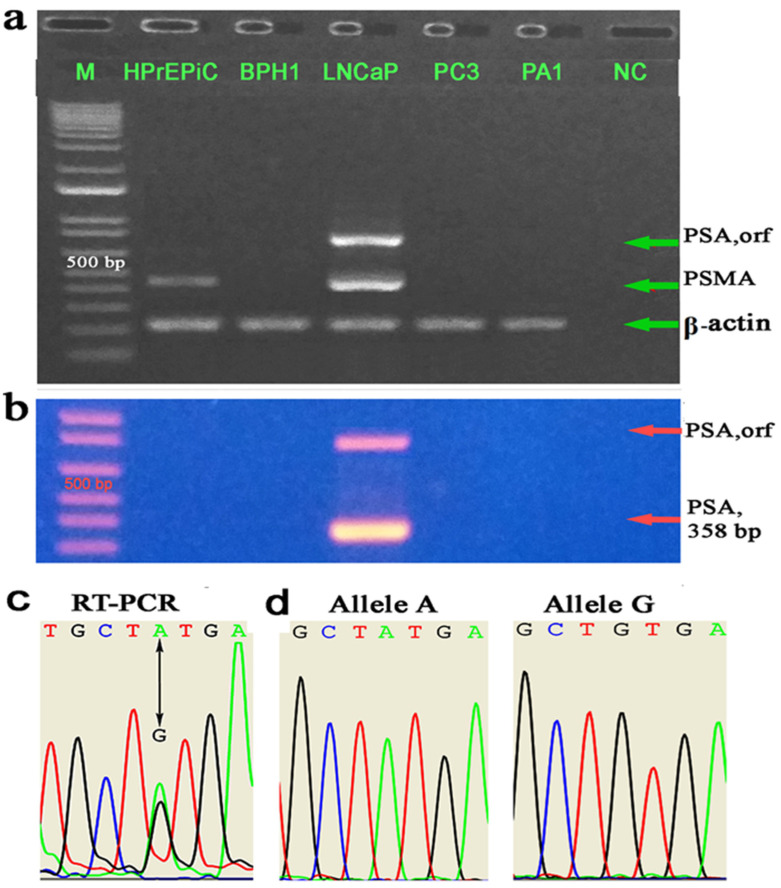
PSA is biallelically expressed in LNCaP cells. (**a**) Gel photograph showing the result of RT-PCR that detects prostate-specific marker expression. (**b**) Gel photograph showing the result of nested PSA RT-PCR. The first round included amplification of the ORF PSA, and the second round used inner primers. (**c**) Chromatogram showing an A/G polymorphism in the coding region of the parental alleles, detected by RT-PCR ORF transcript sequencing, confirming cDNA heterozygosity. Two independent cDNA samples were used in cloning/sequencing experiments for RT-PCR products. (**d**) Chromatograms showing the A or G allele of individual clones that correspond to biallelic (A/G) PSA expression.

**Figure 4 ijms-26-01243-f004:**
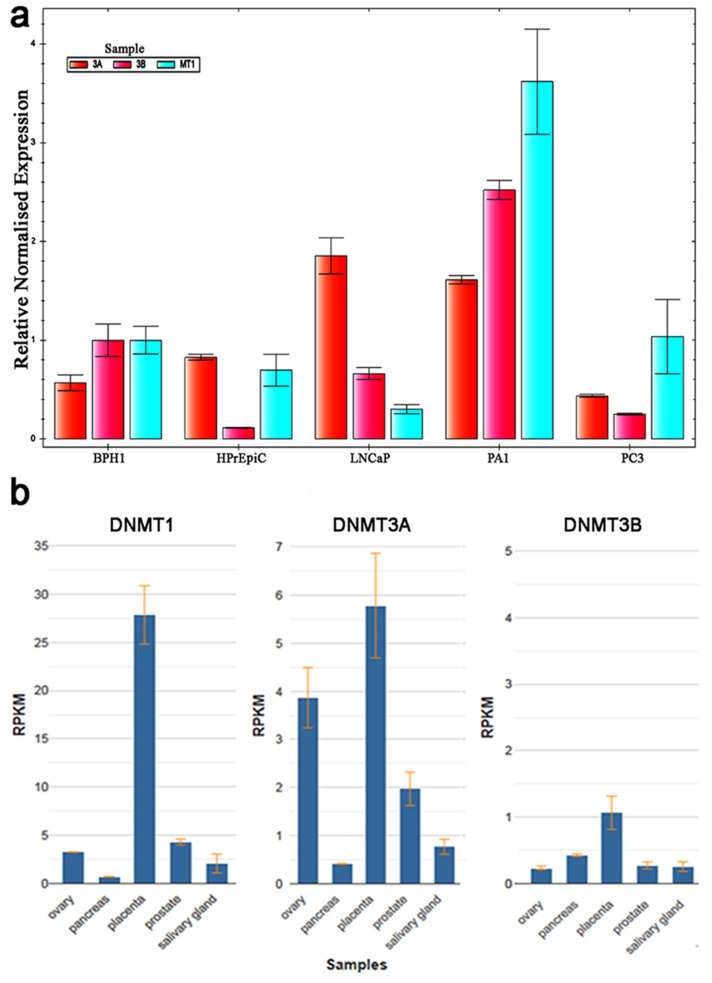
The DNMT profile in cells with PSA monoallelic methylation is similar overall. (**a**) DNMT1 is downregulated in LNCaP cells with an unmethylated PSA promoter. In contrast, it is activated in PC3 cells, where the PSA promoter is biallelically methylated. The relative expression of DNMT1, DNMT3A, and DNMT3B mRNA was assessed by qRT-PCR. The results were normalized to that of TBP expression. All data are presented as the mean ± *SD* of three independent experiments. (**b**) DNMT expression levels determined by RNA sequencing in normal prostate tissues. RNA sequencing was performed on tissue samples from 95 people. Image adapted from NCBI Bioinformatics Database resources: BioProject, PRJEB4337 [13].

**Figure 5 ijms-26-01243-f005:**
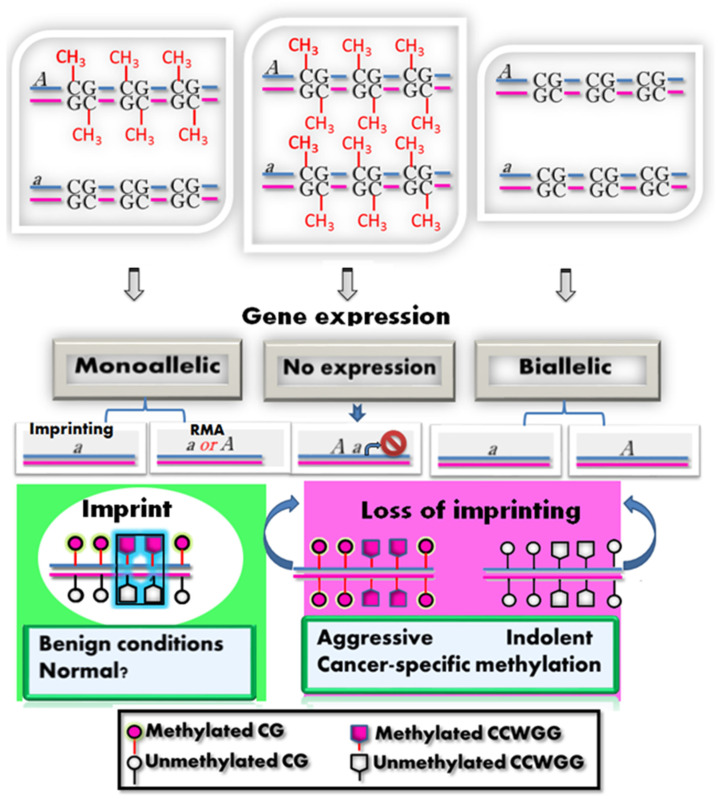
Conceptual diagram of gene expression mediated by methylation. Depending on acquired epigenetic information (methylation state of alleles), gene expression will be realized as follows: Monoallelic expression occurs when only the active allele is transcribed, retaining the unmethylated status. Which of the two alleles is transcribed may depend on each allele’s parental origin, or an allele may be selected at random. When both alleles are no longer methylated, both alleles are transcribed. In the case of biallelic methylation, transcription is abolished. The monoallelic arrangement of CG/CCWGG methylation suggests the ability of the CCWGG mark to serve as a secondary imprint if not present in gDMR, which is known as a primary imprint.

**Table 1 ijms-26-01243-t001:** Primer sequences for RT, RT-qPCR, and bisulfite sequencing analysis.

Gene Symbol	Sequence (5′ ⟶ 3′) Forward/Reverse	Exon Location	Tm (°C)	Size (bp)	Accession No.
DNMT1	GGATTAGGGAGTTTTATAATTTTTTG AACCCACATAATAACACAACTCT	4 6	60	135	NM_001160045
DNMT3A	TATTGATGAGCGCACAAGAGAGC TTGGCACATTCCTCCAACGAAG	11 12	60	136	NM_022552
DNMT3B	GAATTACTCACGCCCCAAGGA TGGCATCAATCATCACTGGATTAC	19 20	60	135	NM_006892
TBP	GAGCCAAGAGTGAAGAACAGTCC AACTTCACATCACAGCTCCCCA	5,6 6	60	130	NM_003194
PSA ORF	AGCTGTGTCACCATGTGGG CTCAGGGGTTGGCCACGA	1 5	60	799	NM_001648
PSA Inner	CAGTCTGCGGCGGTGTTCT GGGTCAAGAACTCCTCTGGTTCA	2 3,4	58	358	NM_001648
PSMA	CAGTCTGCGGCGGTGTTCT CAGGCCAAATTCTTTCCACTGGGA	1 3	60	478	NM_004476
ACTB	CGCCCTGCCTATCTGTATT TCCCCACAGGGAGTGTGTAG	4 5	60	230	NM_001101.5
**Primer sequences for bisulfite sequencing analysis**					
Gene Symbol	Sequence (5′ ⟶ 3′) Forward/Reverse	Promoter Location	Tm (°C)	Size (bp)	Accession No
PSA	GGATTAGGGAGTTTTATAATTTTTTG AACCCACATAATAACACAACTCT	−393 +51	64	443	Acc:HGNC:6364 50,854,915-50,860,764

## Data Availability

The datasets generated and analyzed during the current study are available in the NCBI Sequence Read Archive (SRA) under accession number www.ncbi.nlm.nih.gov/bioproject/PRJNA1007656 (accessed on 12 February 2023).

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
