# Peer review of "Allele-Specific CG/CCWGG Methylation of the PSA Promoter Discriminates Aggressive, Indolent, and Benign Prostate Cell Lines and Is Involved in the Regulation of PSA Expression"

_ijms, 2025, doi:10.3390/ijms26031243_

Round 1

Reviewer 1 Report

Comments and Suggestions for Authors

This manuscript is very interesting and worth publishing, since it highlights a relevant clinical problem for urologists, providing a solution: the difficulty in interpreting a rise in PSA , which can be due to prostate cancer but also, as the Authors well underline, to prostatitis, BPH, prostate traumatism, or others.  The solution proposed by the Authors, is to use the CG/CCWGG methylation in the PSA promoter to increase the diagnostic accuracy of PSA detection. An elevated PSA with monoallelic promoter methylation could indicate a benign condition, while an elevated PSA level due to biallelic PSA expression accompanied by an unmethylated promoter, would suggest prostate cancer. This could have an immediate positive clinical effect, which is a reduction in unnecessary prostate biopsies in patients with elevated PSA. The Authors should add a mention to this important clinical consequence of their work in the Discussion.

In the Introduction, it should be mentioned that PSA is expressed not only by normal, hyperplastic and neoplastic epithelial prostate cells. It is expressed also during prostate inflammation, prostate infection, prostate traumatism, such as irradiation, urethral catheterization, sexual hyperstimulation, or medical procedures, including digital rectal examination. This makes the clinical application of PSA as a reliable marker for prostate cancer detection often very challenging.

Page 12, line 304: In contrast, methylation-based biomarkers are showing highly relevant clinical applications in bladder cancer detection (add citation here:  Mancini M., et al. The Bladder Epicheck Test as a non-Invasive Tool Based on the Identification of DNA Methylation in Bladder Cancer Cells in the Urine: a Review of Published Evidence. IJMS 21, 6542;2020. doi:10.3390/ijms21186542).

Moreover, biallelic methylation-mediated PSA silencing in aggressive prostate cancer could result in low PSA levels in patients, as clinicians know very well. Detection of biallelic methylation even at low PSA levels could increase the possibility of detecting aggressive prostate cancers, increasing diagnostic accuracy. This test could be potentially used to enhance earlier diagnosis in patients harbouring aggressive prostate cancer, even with low PSA or a negative MRI. This important clinical point should be underlined by the Authors in the Discussion section of the paper.

Author Response

Reviewer 1

This manuscript is very interesting and worth publishing, since it highlights a relevant clinical problem for urologists, providing a solution: the difficulty in interpreting a rise in PSA , which can be due to prostate cancer but also, as the Authors well underline, to prostatitis, BPH, prostate traumatism, or others.  The solution proposed by the Authors, is to use the CG/CCWGG methylation in the PSA promoter to increase the diagnostic accuracy of PSA detection. An elevated PSA with monoallelic promoter methylation could indicate a benign condition, while an elevated PSA level due to biallelic PSA expression accompanied by an unmethylated promoter, would suggest prostate cancer. This could have an immediate positive clinical effect, which is a reduction in unnecessary prostate biopsies in patients with elevated PSA. The Authors should add a mention to this important clinical consequence of their work in the Discussion.

In the Introduction, it should be mentioned that PSA is expressed not only by normal, hyperplastic and neoplastic epithelial prostate cells. It is expressed also during prostate inflammation, prostate infection, prostate traumatism, such as irradiation, urethral catheterization, sexual hyperstimulation, or medical procedures, including digital rectal examination. This makes the clinical application of PSA as a reliable marker for prostate cancer detection often very challenging.

Thank you for the suggestion. Since our article is concerned with profiling CG/CCWGG methylation in the PSA promoter in prostate cancer model cell lines, we believe that mentioning a wide range of non-cancer diseases and some medical procedures or even physical activity that may affect blood PSA levels to varying degrees is not so important for the main topic of the article, given that we are dealing with normal prostate cells in all non-cancer diseases.

Page 12, line 304: In contrast, methylation-based biomarkers are showing highly relevant clinical applications in bladder cancer detection (add citation here:  Mancini M., et al. The Bladder Epicheck Test as a non-Invasive Tool Based on the Identification of DNA Methylation in Bladder Cancer Cells in the Urine: a Review of Published Evidence. IJMS 21, 6542;2020. doi:10.3390/ijms21186542).

Page 12, line 304: The monoallelic arrangement of CG/CCWGG methylation suggests the ability of the CCWGG mark to serve as a secondary imprint if not present in gDMR, which is known as a primary imprint.

From a logical point of view, the proposed citation is difficult to mention after the description of the putative involvement of the CG/CCWGG monoallelic location in imprinting. Moreover, it is unknown whether CCWGG methylation exists in bladder cancer.

Moreover, biallelic methylation-mediated PSA silencing in aggressive prostate cancer could result in low PSA levels in patients, as clinicians know very well. Detection of biallelic methylation even at low PSA levels could increase the possibility of detecting aggressive prostate cancers, increasing diagnostic accuracy. This test could be potentially used to enhance earlier diagnosis in patients harbouring aggressive prostate cancer, even with low PSA or a negative MRI. This important clinical point should be underlined by the Authors in the Discussion section of the paper.

The importance of allelic methylation detection for clinical use is clearly highlighted in our paper: lines 321-323

We have modified the last sentence to include some suggestions from the reviewer (marked in yellow)

Reviewer 2 Report

Comments and Suggestions for Authors

The authors describes the role of allele-specific CG/CCWGG methylation of the PSA promoter in PSA expression. The topic is very important to undestand processes in prostate cancers - the article highlights distinct methylation patterns in the PSA promoter that differentiate between aggressive, indolent, and benign prostate states. The article is well written and easy to read and understand. The techniques used are corretc for such kind of studies. The data are correctly done and well described. In my opinion, used 4 cells lines show the correct picture and support the authors' idea. The data are very good discussed based on recent findings in this topic. As for English, I am not native speaker, for me English is acceptable. Everything is clear to read and understand.

I have only minor remarks:

1) The description of statistical methods (p.4.9) is not enough to describe them. Please, read the manual and add the methods of statistical analysis.

2) ScienCell Research Laboratories (city, country)?

3) Other reagents were from Sigma-Aldrich? Fluka? Invitrogen? et cetera.

4) The name of first author and his e-mail name are differet - is it Ok ?

5) References have diferrent styles.

Author Response

Reviewer 2

The authors describes the role of allele-specific CG/CCWGG methylation of the PSA promoter in PSA expression. The topic is very important to undestand processes in prostate cancers - the article highlights distinct methylation patterns in the PSA promoter that differentiate between aggressive, indolent, and benign prostate states. The article is well written and easy to read and understand. The techniques used are corretc for such kind of studies. The data are correctly done and well described. In my opinion, used 4 cells lines show the correct picture and support the authors' idea. The data are very good discussed based on recent findings in this topic. As for English, I am not native speaker, for me English is acceptable. Everything is clear to read and understand.

I have only minor remarks:

  • The description of statistical methods (p.4.9) is not enough to describe them. Please, read the manual and add the methods of statistical analysis.

According to our study, monoallelic CG/CCWGG (BPH1) methylation is a characteristic of non-cancerous conditions in which neither LNCaP allele-specific methylation (indolent PCa phenotype) nor PC3 allele-specific methylations (aggressive PCa phenotype) were observed. These different allelic methylation patterns appear to preclude statistical analysis due to the uniqueness of each. Assessment of CG/CCWGG allelic methylation in clinical samples will be performed using statistical analysis.

  • ScienCell Research Laboratories (city, country)?

Walkersville, MD, USA  has been added.

  • Other reagents were from Sigma-Aldrich? Fluka? Invitrogen? et cetera.

Updated (marked in yellow)

  • The name of first author and his e-mail name are differet - is it Ok ?

OK.

  • References have diferrent styles.

The references style has been updated.